# Evaluation of an mHealth-enabled hierarchical diabetes management intervention in primary care in China (ROADMAP): A cluster randomized trial

**Weiping Jia** [1,2☯]*, **Puhong Zhang** [3,4☯], **Dalong Zhu** [2,5], **Nadila Duolikun** [3], **Hong Li** [6], **Yuqian Bao** [1], **Xian Li** [3,4], **for the ROADMAP Study Group** [¶]

1 Shanghai Jiao Tong University Affiliated Sixth People's Hospital, Department of Endocrinology, Shanghai Diabetes Institute, Shanghai Key Laboratory of Diabetes Mellitus, Shanghai Clinical Center for Diabetes, Shanghai Key Clinical Center for Metabolic Disease, Shanghai, China, 2 Chinese Diabetes Society, Beijing, China, 3 The George Institute for Global Health at Peking University Health Science Center, Beijing, China, 4 Faculty of Medicine, University of New South Wales, Sydney, Australia, 5 Department of Endocrinology, Drum Tower Hospital affiliated to Nanjing University Medical School, Nanjing, China, 6 Department of Endocrinology, Sir Run Run Shaw Hospital affiliated to School of Medicine, Zhejiang University, Hangzhou, China

☯ These authors contributed equally to this work.
¶ Membership of the ROADMAP Study Group is provided in the Acknowledgments.
* wpjia@sjtu.edu.cn

**Data Availability Statement:** Data Availability Statement: All data can be viewed in NODE (http://www.biosino.org/node) by pasting the accession

## Abstract

### Background

Glycemic control remains suboptimal in developing countries due to critical system deficiencies. An innovative mobile health (mHealth)-enabled hierarchical diabetes management intervention was introduced and evaluated in China with the purpose of achieving better control of type 2 diabetes in primary care.

### Methods and findings

A community-based cluster randomized controlled trial was conducted among registered patients with type 2 diabetes in primary care from June 2017 to July 2019. A total of 19,601 participants were recruited from 864 communities (clusters) across 25 provinces in China, and 19,546 completed baseline assessment. Moreover, 576 communities (13,037 participants) were centrally randomized to the intervention and 288 communities (6,509 participants) to usual care. The intervention was centered on a tiered care team–delivered mHealth-mediated service package, initiated by monthly blood glucose monitoring at each structured clinic visit. Capacity building and quarterly performance review strategies upheld the quality of delivered primary care. The primary outcome was control of glycated hemoglobin (HbA1c; <7.0%), assessed at baseline and 12 months. The secondary outcomes include the individual/combined control rates of blood glucose, blood pressure (BP), and low-density lipoprotein cholesterol (LDL-C); changes in levels of HbA1c, BP, LDL-C, fasting blood glucose (FBG), and body weight; and episodes of hypoglycemia. Data were analyzed using

OEP000201 into the text search box or through the URL: http://www.biosino.org/node/project/detail/OEP000201. Researchers can apply for access to anonymized study data and associated documents through NODE. Any cross-border transfer of individual dada must obtain official approval following legal procedures. The export of human-related data is governed by the Ministry of Science and Technology of China (MOST) in accordance with the Regulations of the People's Republic of China on Administration of Human Genetic Resources (State Council No.717).

**Funding:** Funding: This study was funded by the Bethune Charitable Foundation (X-J-2017-014, http://bqejjh.org.cn/) through one donation from Tonghua Dongbao Pharmaceutical Co., Ltd (the ROADMAP study group, PI: WJ). Wuxi BioHermes Bio&Medical Technology Co., Ltd provided point-of-care HbA1c analyzers (A1c EZ 2.0) for universal testing of HbA1c (the ROADMAP study group, PI: WJ). The funders had no role in study design, data collection and analysis, decision to publish, or preparation of the manuscript.

**Competing interests:** Competing interests: I have read the journal's policy and the authors of this manuscript have the following competing interests: WJ is a member of the Editorial Board of PLOS Medicine. The authors declare that no other competing interests exist.

**Abbreviations:** BP, blood pressure; BPHS, basic public health services; CI, confidence interval; CONSORT, Consolidated Standards of Reporting Trials; COVID-19, Coronavirus Disease 2019; DBP, diastolic blood pressure; EOS, end of study; FBG, fasting blood glucose; GEE, generalized estimating equation; HbA1c, glycated hemoglobin; ICC, intraclass correlation coefficient; ITT, intention-to-treat; LDL-C, low-density lipoprotein cholesterol; LMIC, low- and middle-income country; mHealth, mobile health; MI, multiple imputation; OR, odds ratio; ROADMAP, Road to Hierarchical Diabetes Management at Primary Care Settings in China; RD, risk difference; RR, relative risk; SBP, systolic blood pressure; SD, standard deviation.

intention-to-treat (ITT) generalized estimating equation (GEE) models, accounting for clustering and baseline values of the analyzed outcomes. After 1-year follow-up, 17,554 participants (89.8%) completed the end-of-study (EOS) assessment, with 45.1% of them from economically developed areas, 49.9% from urban areas, 60.5 (standard deviation [SD] 8.4) years of age, 41.2% male, 6.0 years of median diabetes duration, HbA1c level of 7.87% (SD 1.92%), and 37.3% with HbA1c <7.0% at baseline. Compared with usual care, the intervention led to an absolute improvement in the HbA1c control rate of 7.0% (95% confidence interval [CI] 4.0% to 10.0%) and a relative improvement of 18.6% (relative risk [RR] 1.186, 95% CI 1.105 to 1.267) and an absolute improvement in the composite ABC control (HbA1c <7.0%, BP <140/80 mm Hg, and LDL-C <2.6 mmol/L) rate of 1.9% (95% CI 0.5 to 3.5) and a relative improvement of 21.8% (RR 1.218, 95% CI 1.062 to 1.395). No difference was found on hypoglycemia episode and weight gain between groups. Study limitations include non-centralized laboratory tests except for HbA1c, and caution should be exercised when extrapolating the findings to patients not registered in primary care system.

## Conclusions

The mHealth-enabled hierarchical diabetes management intervention effectively improved diabetes control in primary care and has the potential to be transferred to other chronic conditions management in similar contexts.

## Trial registration

Chinese Clinical Trial Registry (ChiCTR) IOC-17011325.

Author summary

### Why was this study done?

- In many countries, health outcomes are impacted by the consequences of poor glycemic control among people living with diabetes owing to the low quality of primary care.

- Digital technologies have been increasingly adopted as an innovative solution, especially during the epidemic of communicable diseases like Coronavirus Disease 2019 (COVID-19).

- However, very limited literature reports its clinical effectiveness in primary care settings, and the majority of favorable results are from studies with small sample size.

### What did the researchers do and find?

- A mobile health (mHealth)-based digital platform named Road to Hierarchical Diabetes Management at Primary Care Settings in China (ROADMAP) was designed and tested through a community-based cluster randomized controlled trial that covered 19,546 participants from 864 communities across China.

- Compared with usual care, the intervention lowered the glycated hemoglobin (HbA1c) level by 0.3% and improved the HbA1c control rate by 7.0% absolutely and 18.6% relatively. No additional hypoglycemia episodes and weight gain were found relevant to the intervention.

**What do these findings mean?**

- To our knowledge, this study is the largest in testing the effectiveness and safety of an mHealth-based diabetes management with a diverse coverage of primary care settings.

- The findings indicate that integrated mHealth solution could be effective and safe in improving diabetes control in real life and could be more feasible and acceptable during the pandemic of COVID-19.

- Detailed findings in implementation and evaluation of the intervention could inform the promotion of effective and rational use of limited healthcare resource in China and other low- and middle-income countries (LMICs).

## Introduction

While facing the rising prevalence and growing burden of type 2 diabetes, health systems worldwide, particularly in low- and middle-income countries (LMICs), are also threatened by the consequence of poor glycemic control among people living with diabetes [1–4]. In China, the prevalence of diabetes has reached 11.2%, with an estimated population of 129.8 million [5]. To ensure access to primary care, which is essential for people affected by diabetes and other chronic conditions, a national basic public health services (BPHS) program was introduced in 2009 [6]. The program offers people with diabetes with a minimum of 4 blood glucose tests per year, as well as blood pressure (BP) measurement and lifestyle consultations and medication instructions at each clinic visit, and necessary referral. Diabetes management attained remarkable progress since then, in increased self-awareness (30.1% in 2010, 36.5% in 2013, and 43.3% in 2017) and antihyperglycemic treatment coverage (25.8%, 32.2%, and 49.0%, respectively) [5]. However, the control of glycemic level remains suboptimal with little improvement (49.2% in 2013 and 49.4% in 2017) [5], having only 5.6% of people with type 2 diabetes achieved the composite diabetes control (glycated hemoglobin [HbA1c] <7.0%, BP <130/80 mm Hg, and low-density lipoprotein cholesterol [LDL-C] <2.6 mmol/L) [7]. These indicate strong needs for more effective solution to address poor glycemic control with limited healthcare resource.

The amount and the quality of care are factors affecting the attainment of glycemic control. The BPHS assured that quarterly blood glucose tests was highly unlikely to be sufficient, given the guideline recommended frequency was 2 to 4 times per week for fasting blood glucose (FBG) and postprandial blood glucose testings for patients taking oral antihyperglycemic medication [8]. Critical system deficiencies, including limited and burned-out workforces, unavailable essential medicines and basic technologies, fragmented delivery of care, interoperable information systems, and financial restraints have contributed to the low quality of primary care in preventing and controlling major chronic diseases in developing contexts [4,9–12].

Similar challenges exist in China [13,14]. Recent studies on China's primary care call for quality improving efforts on (1) enhancement of the quality of training for primary healthcare physicians; (2) establishment of performance accountability to incentivize high-quality and high-value care; (3) integration of clinical care with the BPHS; (4) strengthening of the coordination between primary healthcare institutions and hospitals; and (5) building primary healthcare system on digital and innovative technologies [13,15,16].

With the global internet penetration, digital technologies have been increasingly adopted in studies that aim to improve glycemic control [17,18]. However, text messaging is a dominant tool, limited literature reports clinical effectiveness in primary care settings, and the majority of favorable results are from studies with small sample size [18–20]. In addition, while a multitude of mobile applications for type 2 diabetes are available on the market, most focus on self-management or hospital appointments, few attempts to improve the quality of diabetes care in primary care settings by combining enhanced blood glucose monitoring and performance monitoring in a hieratical digital system [21–26].

Therefore, a mobile health (mHealth)-based digital platform named Road to Hierarchical Diabetes Management at Primary Care Settings in China (ROADMAP) was designed and had its effectiveness and feasibility on diabetes control tested through a trial during 2017 to 2019 in diverse primary care settings in China. The hypothesis was that contracted continuing service based on increased monitoring for blood glucose and BP for patients with type 2 diabetes, capacity building, and regular performance evaluation for service providers, facilitated by an mHealth platform, would lead to improved diabetes control. This paper summarizes the main results of the trial.

## Methods

This study is reported in accordance with the Consolidated Standards of Reporting Trials (CONSORT) guidelines for cluster randomized controlled trials (S1 CONSORT Checklist) [27]. Detailed study protocol has been described elsewhere [28]. In brief, we conducted a community-based cluster randomized controlled trial for 12 months to compare the effectiveness of the ROADMAP intervention with usual care.

### Setting and participants

The trial involved a total of 864 communities in 144 counties across 25 eligible provinces in mainland China, covering developed/less developed and urban/rural areas. Within each province, an average of 6 counties and 36 subordinate communities (6 communities from each county) participated in the trial. In each community (cluster), an average of 22 participants were selected at random from a full list of type 2 diabetes patients from local BPHS system. Eligible participants were individuals with diagnosed type 2 diabetes who had registered in the BPHS system; aged 18 to 75 years; residing in the community for the last 6 months with no plan of relocating; and voluntarily participating in the study with informed consent. Patients were excluded if suffering severe physical or psychological problems; unable to attend the site visit; unable to consciously answer questions; women in the process of, or planning for, pregnancy or breastfeeding; or participating in any other clinical trials within the last 6 months.

### Randomization and masking

Randomization was performed centrally upon the completion of baseline data collection. Every 6 communities (i.e., clusters) enrolled by county hospital investigators within same county were randomized into either intervention or usual care group in a 2:1 ratio (i.e., 4 clusters to intervention and 2 to usual care). All participating counties were stratified by the

economic development level of their affiliating provinces and urban/rural locality with the purpose of enrolling equal number of counties for each of the 4 strata. Given the nature of cluster study design, the intervention was implemented as open label after randomization. Assessments, at baseline and end of study (EOS), were carried out by trained local investigators who had no information of group allocation. Before the database was locked, statisticians stayed blinded throughout the study until the statistical analysis plan was finalized without any knowledge of allocation [29].

## Interventions

The multicomponent intervention contained a contracted service package, service providers–targeted capacity building, and performance review for service delivery, which was facilitated through a provider-facing smartphone application Graded ROADMAP and a website. The application connected doctors from primary care clinics within the communities and county hospitals and constituted the hierarchy of care team in the region.

Contracted service package was a set of structured diabetes management services, initiated by blood glucose monitoring and BP measurement at each monthly clinic visit. Fingertip blood glucose monitoring was performed with a Graded ROADMAP–bundled unified meter. A minimum of one fasting and one postprandial blood glucose (on the same day); one BP measurement; one diabetic peripheral neuropathy screening; diet and physical exercise consultation and medication instruction were suggested at each face-to-face visit. The primary care doctors kept track of the results of delivered services with Graded ROADMAP app and took on a role alike gatekeeper by proactively providing patients with routine contacts, monitoring and evaluation, and lifestyle instructions. The app would remind the primary care doctor to request a referral when it captured an indication in patient's record. The referral request then reaches the doctors in the county hospitals with patient's profile and health record for further treatment. County hospital investigators routinely oversaw the implementation of the contracted service following a designated operation manual.

Capacity building consisted of 2 half-day compulsory structured training sessions held at the provincial and county level, using a train-the-trainer approach. The county doctors were trained at provincial level and became the trainer of the county level session. The training materials were developed by the Chinese Diabetes Society based on practicing diabetes guidelines, with the purpose of upskilling service providers by addressing theoretical and operational barriers to type 2 diabetes management and treatment in primary care settings. The structural trainings at provincial and county level were delivered one by one by national and provincial working groups, respectively.

Performance review was calculated in real time based on routinely collected data through Graded ROADMAP, and the results were published on the designated website and quarterly circulated through WeChat (the most popular app for communication, social media, and mobile wallet in China) among the hospitals and communities. The performance assessment was conducted monthly by the national working group. It covered both process and outcome indicators including the frequency of delivered blood glucose and BP measurements, the number of diabetes complication screening, and the number of referrals, as well as the changes in blood glucose/BP levels. Detailed contents and implementation of the intervention was described previously in the published protocol [28].

Within the intervention group, participants were further divided into 2 subgroups based on the utilization of an optional patient-targeting smartphone application (Your Doctor), which supported health education and communication with their contracted doctors. The Your Doctor app was provided to all the participants by their community doctors, but the app use was

based on their willingness and capability in using a smartphone. Participants logged in to this app for no less than 4 times throughout the intervention were defined as active Your Doctor users, while those with less than 4 logins were defined as inactive users. Considering the high penetration of smartphone and mobile internet in China [30], we expected that around half of the participants in the intervention group would be Your Doctor active users.

### Usual care

Usual care were the services provided according to the BPHS.

### Outcomes

The primary outcome was HbA1c control rate (<7.0%; target A) at EOS. The secondary outcomes include the percentage of patients achieving both systolic blood pressure (SBP) <140 mm Hg and diastolic blood pressure (DBP) <80 mm Hg (target B) per Chinese diabetes guideline during the study [31]; percentage of patients achieving LDL-C <2.6 mmol/L (target C); optimal control of composite ABC targets (HbA1c <7.0%, BP <140/80 mm Hg, and LDL-C <2.6 mmol/L; 130/80 mm Hg was also used as alternative BP target to facilitate comparability with other studies); percentage of patients achieving FBG <7.0 mmol/L; changes in levels of HbA1c, BP, LDL-C, FBG, and body weight; and episodes of hypoglycemia (blood glucose <3.9 mmol/L).

HbA1c was obtained from a centrally distributed point-of-care HbA1c analyzer (A1c EZ 2.0) [32]. Standard laboratory tests of blood and urine samples, including fasting glucose, lipid profile, creatinine, and kidney function, were performed by qualified laboratories at county hospitals. Data collection, storage, and reporting followed the "Mobile Application Information Service Regulation" issued by the Cyberspace Administration of China in 2016. Individual participants were encrypted and de-identified. Feedback and queries raised on data integrity, authenticity, and accuracy, as well as schedule management, were achieved using unique identifier reporting.

### Statistical analysis

Detailed sample size calculation methods have been described elsewhere [29]. Briefly, 19,008 eligible patients from 864 communities (22 patients each) would be recruited at baseline, and 16,416 participants (14% drop rate) complete the EOS, which could provide an 89% power to detect a 5% absolute increase in the primary outcome for the intervention group. The sample size calculation assumed that 40% of participants would have well-controlled HbA1c (<7%) at EOS in the control group, with an intraclass correlation coefficient (ICC) of 0.15. All analyses were conducted at the individual level following the intention-to-treat (ITT) principle.

The primary analysis of the intervention effect for primary outcome was conducted using a logistic regression model with generalized estimating equations (GEEs) accounting for clustering within communities and with adjustment of baseline continuous HbA1c. This was the pre-specified alternating model in case of nonconvergence of the log binomial regression [29]. The odds ratio (OR) along with the indirectly derived relative risk (RR) and risk difference (RD) were reported. The RR was indirectly derived from the OR using the following formula: RR = OR ÷ [1 − unadjusted p0 in control × (1 − OR)] [33]. Sensitivity analyses included (1) using SBP <130 mm Hg and DBP <80 mm Hg as target to evaluate BP control and optimal control of composite ABC targets for diabetes patients; (2) further covariate-adjusted analyses; (3) imputed analysis: the multiple imputation (MI) technique using a fully conditional specification was adopted for the imputation. A total of 10 sets of imputed data were created and analyzed using the primary model and then pooled to estimate the treatment effect. When

performing MI, the missing continuous values of HbA1c, FPG, BP, and LDL-C were imputed using linear regression, then dichotomized to get imputed binary outcomes. The variables used in the imputation model included patient characteristics including age, sex, education, economic development level, locality (urban/rural), duration of diabetes, key medical history, and comorbidities; level of HbA1c, FPG, BP, and LDL-C at baseline; level of HbA1c, FPG, BP, and LDL-C at EOS; cluster indicator; and treatment (intervention/control); and (4) subgroup analysis.

Further comparison between active and inactive users of Your Doctor within the intervention group was conducted using an inverse propensity score weighted method based on the primary model. A similar analysis strategy was used for secondary outcomes but replaced with linear regression models with GEE by specifying an identity link for continuous outcomes. The comparison of the number of hypoglycemia episodes was tested using Poisson regression analysis with GEE and with adjustment of baseline episodes of each hypoglycemia. All statistical analyses were done with SAS (SAS Institute, version 9.4).

## Patient and public involvement

Regional investigators and doctors from primary care facilities had input into the study design, the development of the intervention tools, pilot testing phase, and implementation of the intervention through roundtable and periodical national/regional review meetings. Participating doctors were informed of the study progress through monthly newsletters and progress reports. Patients with type 2 diabetes in different areas were interviewed for needs analysis at the preparatory and pilot phases and interviewed again about implementation barriers and facilitators and the burden of their participation as part of a process evaluation at EOS. To encourage active engagement, participants received their results from baseline and EOS assessments. The main results of the study will be disseminated to doctors and participants to boost community involvement in type 2 diabetes management beyond the study.

## Ethics and dissemination

The present study had obtained ethics approval (No: 2016–149) from the Institutional Review Board at Shanghai Sixth People's Hospital, where the leading principal investigator is affiliated with, before the study commenced. Written approval from each participating site was granted by the local hospital research ethics committee and other relevant regional regulatory bodies. All trial participating doctors and patients had provided signed informed consent prior to participant recruitment. Findings from this study will be widely disseminated to participants, academia, and public through peer-reviewed journals, conference presentations, social media, and other applicable mechanisms.

## Results

### Participant allocation and baseline characteristics

Between June 2, 2017 and July 26, 2018, 19,601 eligible patients were recruited from 864 communities. A total of 19,546 completed the baseline assessment and underwent randomization, of whom 6,509 from 288 communities were assigned to usual care and 13,037 from 576 communities to the ROADMAP intervention. Moreover, 17,554 patients (89.8%), 5,794 (89.0%) in control and 11,760 (90.2%) in intervention group, completed the EOS assessment at 1 year of enrollment (Fig 1).

Baseline characteristics were well balanced between 2 groups (Table 1). In general, 45.1% of patients were recruited from economically developed areas, with 49.9% from urban areas, a mean (standard deviation, SD) age of 60.5 (8.4) years, 41.2% male, 6.0 years of median diabetes duration,

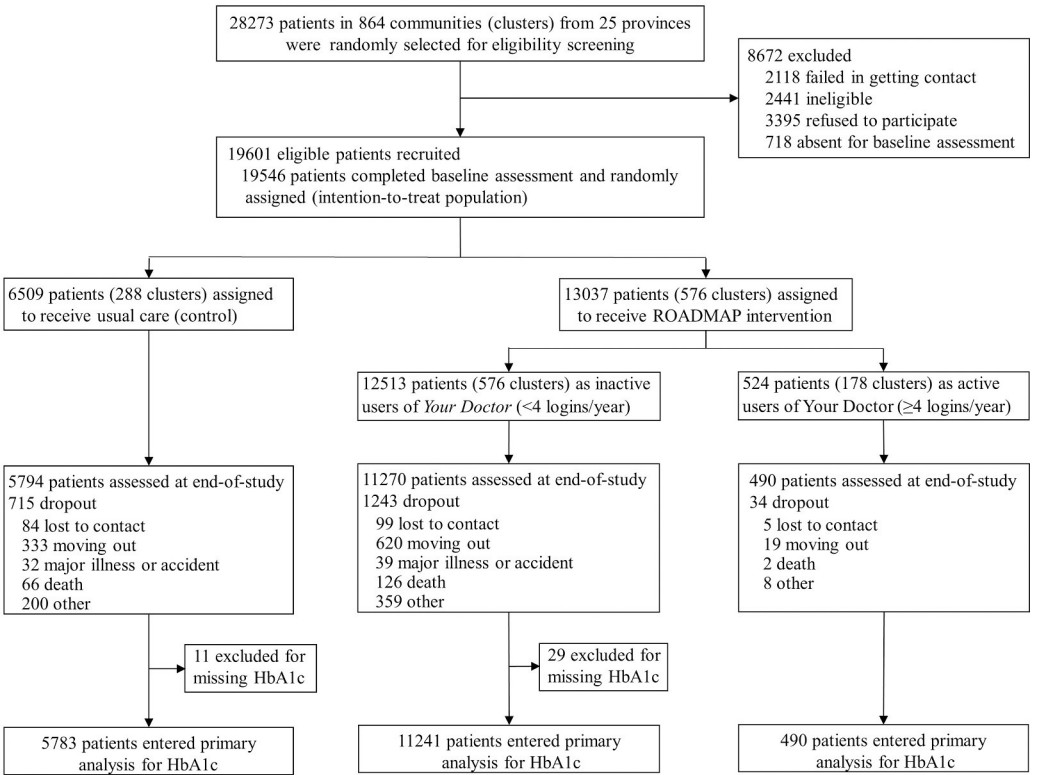

**Fig 1. Participant flowchart.** HbA1c, glycated hemoglobin; ROADMAP, Road to Hierarchical Diabetes Management at Primary Care Settings in China.

mean (SD) HbA1c level of 7.87% (1.92%), 37.3% with HbA1c <7.0%, mean (SD) FBG level of 8.67 (3.46) mmol/L, and 37.1% with FBG<7.0 mmol/L at baseline. The self-reported prevalence for hypertension was 51.8%, dyslipidemia 33.4%, macrovascular disease 18.2%, diabetic retinopathy 11.5%, diabetic nephropathy 7.1%, and diabetic peripheral neuropathy 7.4%. The coefficient of intracluster correlation for the primary outcome (HbA1c level <7.0%) was 0.074.

## Primary and secondary outcomes

Univariate analysis showed that HbA1c control (<7.0%) at EOS was significantly higher in the intervention group (43.6%) compared with the control group (37.5%), an absolute increase of 6.1%. After adjusting for baseline HbA1c and clustering, the primary model showed that the 1-year intervention improved HbA1c control by 18.6% (RR 1.186, 95% confidence interval [CI] 1.105, 1.267) relative to the control group and an absolute increase of 7.0% (95% CI 4.0%, 10.0%). The composite ABC control increased by 21.8% (RR 1.218, 95% CI 1.062, 1.395) relative to the control group and an absolute increase of 1.9% in the primary analysis. Slightly higher control rates were also found for FBG and BP in the intervention group compared with the control group, but the absolute difference or the RRs were not statistically significant. No increase was found for LDL-C (Table 2).

The secondary outcomes measured as continuous variables were presented in Table 3. The multivariable analysis controlling for baseline value of the analyzed outcome and clustering showed that the intervention had an increased effectiveness in lowering the level of HbA1c by 0.30% (95% CI 0.21%, 0.38%), FBG by 0.47 (95% CI 0.32, 0.62) mmol/L, and DBP by 0.6 (95% CI 0.1, 1.1) mm Hg. No significant change was found for SBP and LDL-C.

**Table 1. Baseline characteristics of participants in the ROADMAP study.**

| | Control group (*n* = 6,509) | Intervention group (*n* = 13,037) |
|---|---|---|
| **Region by economic development** | | |
| Developed | 2,935 (45.1%) | 5,877 (45.1%) |
| Less developed | 3,574 (54.9%) | 7,160 (54.9%) |
| **Locality** | | |
| Urban | 3,262 (50.1%) | 6,500 (49.9%) |
| Rural | 3,247 (49.9%) | 6,537 (50.1%) |
| **Demographics** | | |
| Age (years) | 60.8 (8.4) | 60.4 (8.4) |
| Sex (male) | 2,613 (40.1%) | 5,447 (41.8%) |
| BMI (kg/m$^2$) | 25.6 (3.6) | 25.7 (3.5) |
| **Self-reported medical history and complications** | | |
| Duration of diabetes (years), median (Q1, Q3) | 6.0 (3.0, 11.0) | 6.0 (3.0, 10.0) |
| Hypertension | 3,448 (53.0%) | 6,685 (51.3%) |
| Dyslipidemia | 2,158 (33.2%) | 4,376 (33.6%) |
| Any diabetic complication[*] | 2,089 (32.1%) | 4,273 (32.8%) |
| Current smoker | 1,105 (17.0%) | 2,360 (18.1%) |
| **Self-reported examinations (times/person/year)[†]** | | |
| Weight | 4.5 (20.7) | 4.0 (18.7) |
| BP | 21.0 (51.0) | 20.6 (51.8) |
| FBG | 21.4 (39.7) | 21.1 (40.1) |
| Postprandial blood glucose | 10.6 (29.4) | 10.5 (30.1) |
| HbA1c | 0.8 (1.3) | 0.8 (1.3) |
| Blood lipid | 1.0 (1.1) | 0.9 (1.1) |
| Foot: pulsation of the dorsal artery | 0.6 (1.4) | 0.6 (1.3) |
| Neurological examination | 0.2 (0.8) | 0.2 (0.6) |
| **Lab characteristics** | | |
| HbA1c (%) | 7.83 (1.91) | 7.89 (1.93) |
| HbA1c <7% | 2,475 (38.0%) | 4,818 (37.0%) |
| FBG (mmol/L) | 8.57 (3.38) | 8.71 (3.49) |
| FBG <7.0 mmol/L | 2,496 (38.4%) | 4,749 (36.4%) |
| SBP (mm Hg) | 137.1 (21.2) | 136.9 (20. 8) |
| DBP (mm Hg) | 79.7 (12.2) | 79.8 (12.1) |
| LDL-C (mmol/L) | 2.83 (0.91) | 2.84 (0.91) |
| LDL-C <2.6 mmol/L | 2,708 (41.8%) | 5,351 (41.1%) |
| Serum creatinine (umol/L), median (Q1, Q3) | 65.2 (54.5, 79.4) | 66.0 (55.0, 80.0) |

Notes: Data are mean (SD) or *n* (%), unless otherwise specified. Apart from sex, age, and HbA1c level, data were missing for some individuals.

[*] Any diabetic complication is defined as presence of any diagnosed diabetic nephropathy, diabetic retinopathy, peripheral neuropathy, carotid artery disease, lower extremity artery disease, diabetic foot damage, coronary stenosis, myocardial infarction, coronary revascularization, cerebral infarction, or cerebral hemorrhage.

[‡] Self-reported examination data were based on all available sources including tests and measurements conducted at home, primary care clinics, and hospitals.

BP, blood pressure; DBP, diastolic blood pressure; FBG, fasting blood glucose; HbA1c, glycated hemoglobin; LDL-C, low-density lipoprotein cholesterol; ROADMAP, Road to Hierarchical Diabetes Management at Primary Care Settings in China; SBP, systolic blood pressure; SD, standard deviation.

**Table 2. Estimated effects of intervention compared with control on primary and secondary outcomes, with binary outcomes.**

| | Raw data at EOS, *n* (%) | | Effects based on primary model* | | |
| --- | --- | --- | --- | --- | --- |
| | **Control** | **Intervention** | **OR (95% CI)** | **RR (95% CI)** | **RD (%, 95% CI)** |
| **HbA1c <7.0% (primary outcome)** | 2,171 (37.5) | 5,120 (43.6) | 1.335 (1.180, 1.510) | 1.186 (1.105, 1.267) | 7.0 (4.0, 10.0) |
| **FBG <7.0 mmol/L** | 2,357 (40.9) | 5,160 (44.1) | 1.115 (0.993, 1.252) | 1.065 (0.996, 1.135) | 2.7 (−0.2, 5.5) |
| **BP <140/80 mm Hg†** | 2,513 (43.4) | 5,282 (45.0) | 1.055 (0.952, 1.169) | 1.030 (0.972, 1.089) | 1.3 (−1.2, 3.9) |
| **LDL-C <2.6 mmol/L** | 2,615 (45.8) | 5,235 (45.0) | 0.984 (0.865, 1.120) | 0.991 (0.922, 1.062) | −0.4 (−3.6, 2.8) |
| **Composite ABC control‡** | 509 (8.9) | 1,258 (10.9) | 1.245 (1.068, 1.451) | 1.218 (1.062, 1.395) | 1.9 (0.5, 3.5) |

Notes: Composite ABC control is defined as HbA1c level <7.0%, BP <140/80 mm Hg, and LDL-C <2.6 mmol/L.

* Primary model: logistic regression with GEE accounting for clustering and with adjustment of the baseline value of the analyzed outcome. RR are indirectly derived from OR using the following formula: RR = OR ÷ [1 − p0 in control × (1 − OR)].

† Only baseline SBP and clustering were adjusted in the primary model for BP control.

‡ No baseline variable was adjusted in the primary model for the composite ABC control.

BP, blood pressure; CI, confidence interval; EOS, end of study; FBG, fasting blood glucose; GEE, generalized estimating equation; LDL-C, low-density lipoprotein cholesterol; OR, odds ratio; RD, risk difference; RR, relative risk; SBP, systolic blood pressure.

**Table 3. Estimated effects of intervention compared with control on primary and secondary outcomes, with continuous outcomes.**

| | Raw data, mean (SD) | | | | Effects based on primary model* | |
| --- | --- | --- | --- | --- | --- | --- |
| | **Control** | | **Intervention** | | **Difference (95% CI)** | ***P* value** |
| **HbA1c (%)** | | | | | | |
| Baseline | 7.83 | (1.91) | 7.89 | (1.93) | - | - |
| EOS | 7.82 | (1.93) | 7.55 | (1.79) | - | - |
| Change | −0.01 | (1.78) | −0.33 | (1.75) | −0.30 (−0.38, −0.21) | <0.001 |
| **FBG (mmol/L)** | | | | | | |
| Baseline | 8.57 | (3.38) | 8.71 | (3.49) | - | - |
| EOS | 8.50 | (3.51) | 8.08 | (3.14) | - | - |
| Change | −0.06 | (3.44) | −0.63 | (3.49) | −0.47 (−0.62, −0.32) | <0.001 |
| **SBP (mm Hg)** | | | | | | |
| Baseline | 137.1 | (21.2) | 136.9 | (20.8) | - | - |
| EOS | 135.6 | (20.1) | 134.4 | (19.7) | - | - |
| Change | −1.6 | (19.4) | −2.4 | (19.3) | −1.0 (−2.0, 0.0) | 0.052 |
| **DBP (mm Hg)** | | | | | | |
| Baseline | 79.7 | (12.2) | 79.8 | (12.1) | - | - |
| EOS | 79.3 | (11.6) | 78.7 | (11.3) | - | - |
| Change | −0.4 | (11.1) | −1.0 | (11.1) | −0.6 (−1.1, −0.1) | 0.032 |
| **LDL-C (mmol/L)** | | | | | | |
| Baseline | 2.83 | (0.91) | 2.84 | (0.91) | - | - |
| EOS | 2.74 | (0.88) | 2.74 | (0.88) | - | - |
| Change | −0.09 | (0.83) | −0.10 | (0.84) | −0.01 (−0.06, 0.04) | 0.808 |
| **Weight (kg)** | | | | | | |
| Baseline | 64.8 | (11.4) | 64.9 | (11.3) | - | - |
| EOS | 64.5 | (11.5) | 64.6 | (11.2) | - | - |
| Change | −0.2 | (4.5) | −0.2 | (4.3) | 0.0 (−0.2, 0.2) | 0.916 |

* Primary model: linear regression with GEE accounting for clustering and with adjustment of baseline value of the analyzed outcome.

CI, confidence interval; DBP, diastolic blood pressure; EOS, end of study; FBG, fasting blood glucose; GEE, generalized estimating equation; HbA1c, glycated hemoglobin; LDL-C, low-density lipoprotein cholesterol; SBP, systolic blood pressure.

Sensitivity analysis using BP <130/80 mm Hg as target, covariate-adjusted analysis, and imputed analysis regarding the primary and secondary outcomes did not show significant difference when compared with the primary models (S1 Table for binary outcomes and S2 Table for continuous outcomes).

The comparison of the binary outcomes between active and inactive users of Your Doctor among the intervention group is presented in S3 Table. The primary multivariable regression model suggested no significant difference for blood glucose, BP, and composite ABC control between the 2 subpopulations, apart from higher LDL-C control rate in the active users.

### Hypoglycemia and weight gain

With significant improvement in blood glucose control, the intervention did not cause an increase in any type of hypoglycemia (S4 Table) and body weight (Table 3).

### Subgroup analysis

Subgroup analysis illustrated in S1 Fig shows that the difference for HbA1c control between the studied subgroups was not statistically significant, although better HbA1c control was found among participants with age<60 years than age ≥60 years, duration of diabetes ≥6 years than <6 years, from less developed region than developed region, and rural than urban locality.

### Intervention implementation

S5 Table shows that the yearly frequencies of delivered FBG test, postprandial blood glucose test, and BP measurement were 8 to 11 times in the intervention group, which did not achieve the protocol prescribed "monthly" target. A positive association between FBG test frequency and FBG control was detected (S2A Fig), but the trend was not found between BP measurement and BP lowering (S2B Fig). During the intervention period, less than 1 among every 5 patients (0.19/person/year) was successfully referred to upstream doctors, and only 4% of participants in the intervention group were Your Doctor active users (S5 Table).

Diabetes management activities between the 2 groups were compared using patient self-reported data. Participants in the intervention group reported slightly more examinations for blood glucose, BP, HbA1c, and blood lipid than those in the control group. Greater differences were found in mean frequencies of examination for diabetes foot and neurological complication. The improvement in these diabetes management activities in intervention group was also reflexed in the significant increase of total score of SDSCA by 1.2 (95% CI 0.3, 2.0) when compared with the control. The changes in medication were weak in both groups, and a slightly higher use of oral antihyperglycemic drugs was found in the intervention group compared with control (S5 Table).

The performance evaluation report was calculated automatically using the routinely collected data through the ROADMAP platform for intervention group. The algorithms of ranking were based on the monitoring frequency and control rate of the most recent blood glucose and BP as well as patient referral. The results could be assessed anytime by authorized doctors and responsible investigators, and a hard copy will be shared monthly. S6 Table exhibits a real example of the monthly performance report for 24 counties in September 2018.

## Discussion

To our knowledge, ROADMAP is the largest randomized controlled trial in testing the effectiveness and safety of an mHealth-based diabetes management with a diverse coverage of primary care settings. After 1-year follow-up, we found that the intervention lowered the HbA1c

level by 0.3% when compared with usual care, which led to an absolute improvement in the HbA1c control rate of 7.0% (95% CI 4.0% to 10.0%) and a relative improvement of 18.6% (RR 1.186, 95% CI 1.105 to 1.267). We also observed an absolute improvement in the intervention group in the composite cardiometabolic ABC control rate of 1.9% (95% CI 0.5 to 3.5) and a relative improvement of 21.8% (RR 1.218, 95% CI 1.062 to 1.395). The increased control of FBG and BP and the reduced levels of FBG and DBP favored the intervention as well. No increasing episodes of hypoglycemia and weight gain were found to be associated with the intervention. The findings indicate that integrated mHealth solution, which facilitates blood glucose monitoring and performance evaluation, could be effective and safe in improving diabetes control in primary care settings.

Regular blood glucose monitoring is the basis for optimized diabetes management. The minimum requirement of 4 blood glucose tests for each patient per year by the BPHS in China has been an improvement in universal coverage of diabetes management, but it is far from enough. Our study showed that a slight increase in the frequency of blood glucose monitoring was accompanied by other behavior change and better glycemic control. Studies from Europe also demonstrated that app-supported blood glucose monitoring could improve HbA1c and had the potential to reduce the cost [34,35]. A fixed 2 tests of blood glucose per month was required in this study, but a varied frequency of monitoring based on individualized goals and willingness has been suggested by other study [36].

Persistent uncontrolled glycemic level is one of the referral indications as per the Chinese diabetes guideline. However, the average referral frequency found in the intervention group of the ROADMAP study was only 0.19 time per patient per year. This is disproportionately low considering nearly 60% participants did not achieve the HbA1c target (<7%). In addition to emphasizing the importance of intensive treatment toward the target, appropriate patient referral to/from upstream hospitals should be strengthened. These results suggested that purely technological support was insufficient, and other measures including strict pathway or regulation with clear responsibilities and result-oriented budgeting should be considered.

Previous studies have yielded the promising potential of applying digital health tools in diabetes management. A meta-analysis examined 181 trials assessing diabetes management interventions, which involved multiple players including health system, healthcare providers, and patients [37]. The pooled result showed a mean HbA1c reduction by 0.28% (95% CI: 0.21, 0.35), ranging from −1.80% [38] to 0.70% [39]. Among them, interventions based on digital health such as electronic patient registry [40], electronic decision-making [41], and information relay from patients to clinicians demonstrated on average a 0.28% to 0.36% reduction in HbA1c level [42], which is consistent with the findings from our study. The trials that reported an improvement of over absolute 1% HbA1c were mostly those adopting active pharmacist engagement [43,44] or interactive workshops for patient education [38,45], indicating that strengthened functionalities on medication and patient education should be included in the ROADMAP platform in the future.

Despite the variability in intervention strategies, the majority of studies reporting positive effect were dominated by trials with small to medium sample sizes, while the trials with larger populations tended to be negative [18], particularly for those implemented among patients broadly recruited from communities [46]. Acknowledging the difficulties in implementing complex intervention in diverse context, we adopted several strategies to increase intervention fidelity, including clarifying intervention flowchart using table stickers, providing remote trainings and reminders through WeChat, and real-time access to performance report and ranking through a designated website portal.

Even with above strategies, the intervention delivery remained inadequate. Based on the app records, the frequency of blood glucose test was 18.7 times/year, less than the required 24

times/year; the frequency of BP measurement was 8.6 times/year, less than the required 12 times/year; the referral (0.19 times/year) was also significantly underused as explained above; and only 4% of participants in intervention group positively used Your Doctor to communicate with and receive education from their doctors, although we expected the number was around 50%. Possible reasons reported by other studies for low fidelity in implementing a complex intervention include the need of longer learning time [47–49], competing workload [50], and not user-friendly for elderly users [23]. Drawn from our process evaluation, however, fidelity issues were largely driven by low expectation of treat to target for blood glucose/BP by both patients and service providers, which resulted in negligence of alerts to abnormal blood glucose/BP measurements and referral indications by service providers and low patient compliance to instructions given by the providers.

The ROADMAP study presents several strengths. First, this study has a diverse coverage of 864 communities across 25 Chinese provinces. It provides an opportunity to explore context impact and make context-fitting adaptation for future scale-up. Secondly, the participants in the ROADMAP study were randomly selected from all patients registered in the studied communities. The representative sampling enhances the generalizability of the findings. Thirdly, a designated mobile phone–based electronic data collection application base on our past trial experiences [51] was adopted to support the project management and data collection, which largely increased the quality and efficiency of project implementation and quality. Furthermore, the centrally distributed HbA1c analyzers for point-of-care testing guarantees the accuracy and consistency of the primary outcome throughout such a large study.

There are several limitations in the ROADMAP study. The participants were recruited from registered patients with diabetes; caution is suggested when extrapolating the findings to a general population with type 2 diabetes. Apart from HbA1c, other physical and biochemical examinations were conducted using various local resources, although quality requirements had been met during site selection. Third, we underestimated the frequency of blood glucose monitoring in usual care, which resulted in a relatively small improvement in frequency of blood glucose monitoring in our study. A larger effectiveness on diabetes control might be achieved if more frequent individualized blood glucose and BP tests were adopted. Finally, only 4% of participants in the intervention group were Your Doctor active users. Potential bias and low statistical power made it unreliable to detect the effectiveness of Your Doctor.

In conclusion, the primary care system in China, as in other LMICs, is struggling to meet the gaps in chronic disease management, including diabetes, and innovative solutions are needed. The ROADMAP study offers a supportive and collaborative solution to tackle the suboptimal diabetes control in primary care via an integrated mHealth-based platform. More interactive health education to raise the awareness of the importance of treat to target for both patients and community doctors, more close engagement of upstream hospital doctors, leveraging home-tested glucose data to support diabetes management, and adding individualized medication guide to the mHealth platform may further improve the effectiveness of such an mHealth platform. We expect that the findings from the ROADMAP study could inform the improvement of the BPHS programs and promote effective and rational use of limited healthcare resource in China and other LMICs.

## Supporting information

**S1 CONSORT Checklist. CONSORT 2010 checklist of information to include when reporting a cluster randomized trial.** CONSORT, Consolidated Standards of Reporting

Trials.
(DOCX)

**S1 Fig. Forest of subgroup analysis for primary outcome—HbA1c <7% at EOS.** EOS, end of study; HbA1c, glycated hemoglobin.
(PDF)

**S2 Fig. The relationship between frequency of BP/FBG monitoring and BP/FBG control across 12 months in intervention group of ROADMAP study.** BP, blood pressure; FBG, fasting blood glucose; ROADMAP, Road to Hierarchical Diabetes Management at Primary Care Settings in China.
(PDF)

**S1 Table. Sensitivity analysis for estimated effect (binary outcomes) of intervention compared with control.**
(DOCX)

**S2 Table. Sensitivity analysis for estimated effect (continuous outcomes) of intervention compared with control.**
(DOCX)

**S3 Table. Comparison of primary and secondary binary outcomes between Your Doctor inactive and active users within intervention group.**
(DOCX)

**S4 Table. Number of patients and episodes of hypoglycemia within 1 month before EOS.** EOS, end of study.
(DOCX)

**S5 Table. Comparison of key diabetes management activities between 2 groups during 1-year follow-up in the ROADMAP study.** ROADMAP, Road to Hierarchical Diabetes Management at Primary Care Settings in China.
(DOCX)

**S6 Table. Monthly performance of intervention implementation in the ROADMAP study —results of 24 out of 144 counties in September 2018 as an example.** ROADMAP, Road to Hierarchical Diabetes Management at Primary Care Settings in China.
(DOCX)

## Acknowledgments

The authors acknowledge personnel who engaged in helping us accomplish all trial procedures. Appreciation for the support from trial steering committee and efforts made by the ROADMAP Study Group, all participating clinical research associates, investigators, doctors, and patients, as well as those who helped to facilitate trial preparation, execution, and evaluation. The ROADMAP Study Group was composed by Dalong Zhu, Dongmei Li, Guangyao Song, Haoming Tian, Hong Li (Zhejiang Province), Hong Li (Yunnan Province), Hongyu Kuang, Huili Zhang, Jianying Liu, Jie Liu, Jing Liu, Li Chen, Li Yuan, Liming Chen, Lixin Shi, Minxiang Lei, Ping Liu, Puhong Zhang, Qifu Li, Qiu Zhang, Qiuhe Ji, Yadong Sun, Yanbing Li, Yuqian Bao, Yuzhen Liang, Weiping Jia, Zhigang Zhao, Zhongyan Shan, and Zilin Sun, alphabetically ordered by given name. We thank Professor Craig Anderson for providing constructive guidance in writing this manuscript.

## Transparency statement

The corresponding author affirms that the manuscript is an honest, accurate, and transparent account of the study being reported; that no important aspects of the study have been omitted; and that any discrepancies from the study as planned have been explained.

## Author Contributions

**Conceptualization:** Weiping Jia, Puhong Zhang.

**Data curation:** Puhong Zhang, Nadila Duolikun, Xian Li.

**Formal analysis:** Puhong Zhang, Xian Li.

**Funding acquisition:** Weiping Jia.

**Investigation:** Weiping Jia, Puhong Zhang, Dalong Zhu, Nadila Duolikun, Hong Li, Yuqian Bao.

**Methodology:** Weiping Jia, Puhong Zhang, Xian Li.

**Project administration:** Weiping Jia, Puhong Zhang, Nadila Duolikun, Hong Li, Yuqian Bao.

**Supervision:** Weiping Jia, Puhong Zhang, Dalong Zhu.

**Writing – original draft:** Nadila Duolikun.

**Writing – review & editing:** Weiping Jia, Puhong Zhang, Dalong Zhu, Nadila Duolikun, Hong Li, Yuqian Bao, Xian Li.

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
