## [Editor Report · Decision Letter 0]

20 Jan 2021

Dear Dr Jia, 

Thank you for submitting your manuscript entitled "Road to hierarchical diabetes management at primary care (ROADMAP) in China: main results of a cluster-randomized controlled trial" for consideration by PLOS Medicine.

Your manuscript has now been evaluated by the PLOS Medicine editorial staff and I am writing to let you know that we would like to send your submission out for external assessment.

Once your full submission is complete, your paper will undergo a series of checks in preparation for external assessment

Kind regards,

Richard Turner, PhD

rturner@plos.org

---

## [Decision Letter · Decision Letter 1]

15 Mar 2021

Dear Dr. Jia,

Thank you very much for submitting your manuscript "Road to hierarchical diabetes management at primary care (ROADMAP) in China: main results of a cluster-randomized controlled trial" (PMEDICINE-D-21-00273R1) for consideration at PLOS Medicine. 

Your paper was evaluated by an academic editor with relevant expertise and sent to independent reviewers, including a statistical reviewer. The reviews are appended at the bottom of this email and any accompanying reviewer attachments can be seen via the link below:

[LINK]

In light of these reviews, we will not be able to accept the manuscript for publication in the journal in its current form, but we would like to invite you to submit a revised version that addresses the reviewers' and editors' comments fully. You will appreciate that we cannot make a decision about publication until we have seen the revised manuscript and your response, and we expect to seek re-review by one or more of the reviewers. In this case we may need to enlist one or more additional reviewers. 

We hope to receive your revised manuscript by Apr 05 2021 11:59PM. Please email us (plosmedicine@plos.org) if you have any questions or concerns.

Please let me know if you have any questions, and we look forward to receiving your revised manuscript. 

Sincerely,

Richard Turner, PhD

rturner@plos.org

Noting PLOS' data policy (https://journals.plos.org/plosmedicine/s/data-availability) please adapt your data statement to note that data will be made available from the data of publication. Please remove the sentence "... if the proposed use of data is adjudicated adequate to achieve the purpose.".

Please adapt the title to better match journal style. We suggest "Evaluation of an mHealth-enabled hierarchical diabetes management intervention in primary care in China (ROADMAP): A cluster-randomized trial".

Please adapt the abstract to a three-part structure. Please add a new final sentence to the "Methods and findings" subsection, which should begin "Study limitations include ..." or similar and quote 2-3 of the study's main limitations. 

Please quote the study dates in the abstract. We suspect that "1717,754" contains an error - please correct. 

Please quote aggregate demographic details for study participants in your abstract.

Please list secondary outcome measures in your abstract as appropriate. 

Please make that "data were" in the abstract and any other instances. 

After the abstract we will need to ask you to add a new and accessible "Author summary" section in non-identical prose. You may find it helpful to consult one or two recent research papers in PLOS Medicine to get a sense of the preferred style. 

Please move the ethics information from the end of the text to the Methods section. 

Throughout the text, please adapt reference call-outs to the following style: "... exist in China [13,14]." (noting the absence of spaces within the square brackets). 

Please remove the information on funding, competing interests and data sharing from the end of the main text. In the event of publication, this information will appear in the article metadata, via entries in the submission form. 

Throughout the paper, please quote exact p values or "p<0.001".

Please revisit the reference list to ensure that all citations match journal style. All italics should be converted into plain text; and 6 author names should be listed, where appropriate, rather than 3, followed by "et al.".

Please rename figure 1 "Participant flowchart" or similar. 

Please attach a completed CONSORT checklist, labelled "S1_CONSORT_Checklist" or similar and referred to by this label in your methods section. In the checklist, please refer to individual items by section (e.g., "Methods") and paragraph number rather than by line or page numbers, as the latter generally change in the event of publication. 

Comments from the reviewers:

*** Reviewer #1: 

[See attachment]

Michael Dewey

*** Reviewer #2: 

This is a very interesting paper. This outbreak of covid-19 is accelerating the progress of the development and promotion of the mHealth system, but there is no such large sample size of clinical research data in mHealth at present. This study can be seem to the international initiative and has great social effect, because the mHealth management caver in people with different living habits across regions and climates, and it also shows the ideal effect of blood glucose management. Therefore, it can be considered that the results of this study are acceptable to reflect the blood glucose management effect of mHealth system in real life. The grouping design, reasonable intervention measures and statistical results are highly reliable. Therefore, I suggest that it be published in the journal.

*** Reviewer #3: 

By Mihiretu M. Kebede (PhD)

This is a very well-written manuscript. I have few points for consideration. 

Methods: the authors mentioned they have imputed data. Clarify the imputation methodology. 

The intervention is not adequately described. 

Fig1 (trial profile): Both "Your doctor active" and "Your doctor inactive" show similar login values (>=4 logins per year)

Majority of the participants are "Your doctor in active" users. Can you give a clear context why this happened? Highlight whether the active/inactive status makes a difference in the effects of the intervention. 

I was expecting the absolute HbA1c change to be one of the key outcomes. HbA1c change of 0.3 is clinically significant. This needs to be highlighted in the results section as one of the key results. 

Supplementary table shows diabetes self-care activities significantly gets improved. Please highlight that in the results section. 

There are some typos across the manuscript. Please correct

Abstract: Results section, please correct 1717, 754. 

Abbreviation ABC is not clear. 

Change Table 21 to Table 1

***

[LINK]

---

## [Decision Letter · Decision Letter 2]

28 Jul 2021

Dear Dr. Jia,

Thank you very much for re-submitting your manuscript "Evaluation of an mHealth-enabled hierarchical diabetes management intervention in primary care in China (ROADMAP): A cluster-randomized trial" (PMEDICINE-D-21-00273R2) for consideration at PLOS Medicine. We apologize for the delay in sending you a response. 

I have discussed the paper with our academic editor and it was also seen again by one reviewer. I am pleased to tell you that, provided the remaining editorial and production issues are fully dealt with, we expect to be able to accept the paper for publication in the journal.

[LINK]

Please let me know if you have any questions, and we look forward to receiving the revised manuscript.   

Sincerely,

Richard Turner, PhD

rturner@plos.org

Requests from Editors:

Please adapt the information on seeking access to study data in the data statement, as "... following the instruction of NODE" may not be clear to some readers: would "Researchers can apply for access to anonymized study data and associated documents through NODE" be suitable wording?

Please adapt the short title to "Cluster-randomized trial of diabetes management in China" or similar.

In the abstract, we suggest adapting "... data were primarily analyzed" to "... data were analyzed". 

Where you discuss the primary findings in the abstract, please adapt the phrasing to "... the intervention led to an absolute improvement in the HbA1c control rate of 7% (95% CI ...) and a relative improvement of 18.6% ... " or similar. 

In the final sentence of the "Methods and findings" subsection of your abstract, please amend the text to " ... and caution should be exercised when extrapolating ...".

In the Author summary, we suggest amending the first point to "In many countries, health outcomes are impacted by the consequences of poor glycemic control among people living with diabetes owing to the low quality of primary care." or similar. 

Later on in this section, please make that "To our knowledge, this study is the largest in testing ..." or similar.

Please restructure the Results section so that the presentation of primary and secondary endpoint findings comes before the "Intervention implementation" subsection. 

Where you present the primary findings, at the start of p.17 of the PDF, you mention "... an absolute reduction of 7.0% (95% CI 4.0 - 10%)." Should this be an "absolute increase"?

Immediately before table 2 you quote an "absolute increase of 2.0%" for composite ABC control, whereas the table appears to quote this as 1.9%: please check that these are consistent.

In the first paragraph of the Discussion, please quote the trial's primary endpoint findings before the discussion of the intervention components such as "Your Doctor". 

In the final paragraph of the Discussion, please amend the first sentence to "In conclusion, the primary care system in China, as in other LMIC, is struggling ...". 

Throughout the text, please remove spaces from the reference call-outs (e.g., "... innovative technologies [13,15,16].").

Please remove the information on competing interests from references 3, 5 and any other relevant references. 

Comments from Reviewers:

*** Reviewer #1: 

The authors have addressed my points

Michael Dewey

***

[LINK]

---

## [Editor Report · Decision Letter 3]

4 Aug 2021

Dear Dr Jia, 

On behalf of my colleagues and the Academic Editor, Dr Basu, I am pleased to inform you that we have agreed to publish your manuscript "Evaluation of an mHealth-enabled hierarchical diabetes management intervention in primary care in China (ROADMAP): A cluster-randomized trial" (PMEDICINE-D-21-00273R3) in PLOS Medicine.

PRESS

Sincerely, 

Richard Turner, PhD 

rturner@plos.org